# A NON-LINEAR THEORY FOR SENTENCE EMBEDDING

## ABSTRACT

This paper revisits the Random Walk model for sentence embedding in the context of non-extensive statistics. We propose a non-extensive algebra to compute the discourse vector. We argue that by doing so we are taking into account high non-linearity in the semantic space. Furthermore, we show that by considering a non-extensive algebra, the compounding effect of the vector length is mitigated. Overall, we show that the proposed model leads to good sentence embedding. We evaluate the embedding method on textual similarity tasks.

## 1 INTRODUCTION

Word embeddings has become an ubiquitous choice for the word representation block of any natural language processing system. These representations are fixed-length vectors obtained by training predictive models such as Glove (Pennington et al., 2014a) and word2vec (Mikolov et al., 2013) to unlabeled text corpora. Word vectors not only can be used to find similarity between words or to explore analogies, when pretrained on large corpora they can be plugged into a variety of downstream tasks to improve their performance by incorporating some general knowledge learned on the large dataset. While Word2Vec builds the vectorial representations of words by learning the co-occurrence of words in different contexts, fastText enriches the word vectors by breaking them down to n-gram units and as a result is able to predict vectors for out-of-vocabulary words. Deep conceptualized word representation (ELMO) (Peters et al., 2018) has been introduced recently which combines the internal states of a pretrained deep bidirectional language model (biLSTM) as representation of words. ELMO is character-based and models both syntactic and semantic characteristics of words across linguistic contexts.

However, in most of the NLP tasks we are interested in finding an effective representation of sentences or pieces of text not only words. RNNs (Cho et al., 2014) and convolutional architecture with attention mechanisms (Vaswani et al., 2017) are well known models for learning a representation of word sequences and have been investigated in generating sentence embeddings that can be transferred to other tasks (Li et al., 2015; Palangi et al., 2015; Kiros et al., 2015). Deeper models such as universal sentence encoder (Cer et al., 2018) reach high performance with the cost of high memory and compute resources. Infersent (Conneau et al., 2017) improves sentence embeddings by taking advantage of supervision and with the cost of annotation.

Although neural network based models perform well when trained and used in same domain, as shown in (Wieting et al., 2016), when it comes to transferable sentence embedding a simple averaging scheme can out-perform LSTMs. Arora (Arora et al., 2017) showed that a weighted averaging with mild denoising based on singular vectors outperforms DNN models such as skipthought, introduced in (Kiros et al., 2015), in certain downstream tasks. An even stronger deep-learning-free sentence embedding is proposed in (Arora et al., 2018), where order of n-grams is incorporated to enrich the linear combination of word embeddings and obtain a sentence embedding that has the information of local order in sentences. In (Ethayarajh, 2018) the probability of word generation is taken as inversely related to the angular distance between the word and sentence embeddings, which improves Arora et al.'s (Arora et al., 2017) by up to 44.4% on textual similarity tasks. The idea of deep-learning-free sentence embedding is appealing since it is interpretable, unsupervised and simple in terms of computation cost and implementation complexity.

This work is based on the random walk model (Arora et al., 2017). We modify the model to consider the case of non-extensive statistics.

## 2 BACKGROUND

### 2.1 THE RANDOM WALK MODEL

We briefly recall the log linear random walk proposed by (Arora et al., 2016). In this model, words are generated dynamically by a random walk of a time variant discourse vector $c_t \in \Re^d$, representing "what is being talked about". The probability of a word $w$, represented by a vector $v_w \in \Re^d$, being generated at time $t$ is given by a log-linear production model (Mnih & Hinton, 2007):

$$p(w|c_t) \propto exp(< c_t, v_w >) \tag{1}$$

Arora et al. made the assumption of a static discourse vector over a sentence and dropped the time dependence. Therefore, he replaced the sequence of discourse vectors $c_t$ across all time steps with a single discourse vector $c_s$. The MAP estimate of $c_s$ is then the unweighted average of word vectors (ignoring any scalar multiplication).

An improved version of the random walk model was proposed in (Arora et al., 2017), where the words are allowed to be generated by three different mechanisms; (a) by chance with a probability $\alpha p(w)$, where $\alpha$ is some scalar and $p(w)$ is the frequency of a word $w$ in a given corpus; (b) if the word is correlated with the discourse vector $c_t$ and the inner product is not null i.e., $< v_w, c_t > \neq 0$; (c) if the word is correlated to the common discourse vector $c_0$, related to syntax (such as occurrences of stop words). This modification in the random walk model helps to explain the occurrence of words that are weakly correlated to the discourse vector, such as "and", "a", and "the" etc. One can then write the probability of a word $w$ to be generated by a discourse vector $\tilde{c}_s$ as:

$$p(w|c_s) = \alpha p(w) + (1 - \alpha) \frac{exp(< \tilde{c}_s.v_w >)}{Z_{\tilde{c}_s}}, \tag{2}$$

where $\tilde{c}_s \triangleq \beta c_0 + (1 - \beta)c_s$ with $c_0 \perp c_s$ and $Z_{\tilde{c}_s} \triangleq \Sigma_{w' \in \mathbf{V}} exp(< \tilde{c}_s, v_w >)$.

Here, $\alpha$ is a scalar hyperparameter, $\tilde{c}_s$ is a linear combination of the discourse vector and common discourse vectors parameterized by $\beta$, and $Z_{\tilde{c}_s}$ is the partition function.

The vector representation of a sentence is defined as the *MAP* estimate of the discourse vector $c_s$ that generated the sentence. Arora et al. assume that the word vectors are uniformly dispersed in the latent space. This assumption implies that the partition function $Z_{\tilde{c}_s}$ could be assumed to be the same for all $\tilde{c}_s$. Therefore, the partition function could be replaced with a constant $Z$. Assuming a uniform prior over $\tilde{c}_s$, Arora et al. derived the maximum likelihood estimator for $\tilde{c}_s$ on the unit sphere (ignoring normalization) as being proportional to $\frac{1}{|s|} \Sigma_{w \in s} \frac{a}{a+p(w)}.v_w$, where $a \triangleq \frac{1-\alpha}{\alpha.Z}$.

The resulting weighting scheme is called *smoothed inverse frequency* (SIF). Since the weights are inversely proportional to the frequency of the word $w$. The common discourse vector $c_0$ is estimated from the first principal component of all $\tilde{c}_s$ in the corpus. The final discourse vector $c_s$ is then computed by subtracting the projection of the weighted average on the common component:

$$c_s \triangleq \tilde{c}_s - proj_{c_0} \tilde{c}_s \tag{3}$$

### 2.2 CONFOUNDING EFFECT OF VECTOR LENGTH

In the present section, we explain the confounding effect of vector length. This point has been addressed by (Ethayarajh, 2018) in the same context using a different approach. We show that there is no need to drop the log-linear assumption in the definition of the likelihood.

Consider a pair of rare words $x$ and $y$ in a sentence and assume that $x, y \in g$. Let us assume that the two words are not similar i.e., $< v_x, v_y >= 0$. Also let us consider a sentence h where a given word, z, appears two times. Furthermore, let us assume that the removal of common components has negligible effect:

$$c_g = \tilde{c}_g = \frac{1}{2}(\frac{a}{a + p(x)}.v_x + \frac{a}{a + p(y)}.v_y)$$
$$c_h = \tilde{c}_h = \frac{a}{a + p(z)}.v_z \tag{4}$$

Another assumption that we make is that the words $x$,$y$,$z$ are so rare that the probability of them being emitted by chance via the term $p(w)$ is negligible. Therefore, the probability of the words to

be emitted is proportional to the inner product of its vector with the discourse vector. Given that the discourse vector of the sentence $h$ is identical to the word $z$ and given that $x$, $y \in g$ have no similarity, it follows that we would expect:

$$p(h|c_h) > p(g|c_g). \tag{5}$$

However this is not always true. Let us assume the word representation lie in $\Re^2$. Let us rewrite the vectors $x$, $y$, $z$:

$$v_x = 2ke_x, v_y = 2ke_y, v_z = ke_x + ke_y. \tag{6}$$

We first assume that the words $x$, $y$, and $z$ are rare and the probabilities $p(x) \approx p(y) \approx p(z) \approx 0$. Furthermore, let assume that the common components related to syntax are negligible and that $\tilde{c}_s \approx c_s$. In this case, the probabilities $p(h|c_h)$ and $p(g|c_g)$ are equal:

$$c_g = c_h = \frac{a}{a + p(x)}(ke_x + ke_y)$$
$$\implies <c_g, x> = <c_g, y> = <c_g, z> = 2k^2 \tag{7}$$
$$\implies p(g|c_g) = p(h|c_h).$$

This is in contradiction with 5. In this particular case, this contradiction is mainly due to the fact that the probability of occurrence of a word depends on the inner product of the word and discourse vector and that the discourse vector is computed as the average of the words in the sentence. Linearity is assumed in the estimation of the discourse vector $\tilde{c}_s$ i.e., we approximate $\tilde{c}_s$ to the linear average of the words in the sentence. In the following sections, we show that by taking into account non-linearity we can lift the confounding effect of the vector length.

## 2.3 Non-extensive statistics

Non extensive statistics originates from the study of thermodynamical systems with long range interactions where thermodynamic quantities are not proportional to the volume or to the number of particles in the given system (Tsallis, 1988; 2008). In the field of physics, theoretical work has derived many models characterized by non-additive energy or entropy, e.g., black holes and some magnetic and fluid models characterized by complex and long range interactions. In this context, the probability density function characterizing the state of the system is no longer dictated by the Boltzmann-Gibbs statistics and the distribution considered are non-Gaussian distributions. More particularly, the generalized distribution function called q-exponential, given by $exp_q(x) = [1 + (1-q)x]^{\frac{1}{1-q}}$, where $q$ is a parameter indicating degree of non-extensivity of the system, is considered.

Although originating from physics, these functions present important mathematical interest. They have been studied from a mathematical point of view and more particulary for algebra generalization (Yamano, 2002; Aguiar & Kodama, 2003; Borges, 2004).

In the present work, we test the hypothesis of non-extensivity of the discourse in the semantic space and derive a new generative model based on non-extensive statistics.

## 3 Non-extensive Random Walk Model

We argue that due to the complexity of natural language, non-linearity in the semantic space should be taken into account. More particularly, the discourse vector $\tilde{c}_s$ should not be approximated to the linear average of the word vectors, but should include some non-linearity from the semantic space. In this context, we propose the use of non-extensive statistics as a framework.

In order to generalize the log-linear function proposed by (Mnih & Hinton, 2007), we want to find a class of functions $F$ that satisfy non-extensivity conditions in the semantic space. For this purpose, let us assume that $F$ is a morphism between the groups $(D_q, +)$ and $(R_{>0}, \otimes_q)$:

$$F(\vec{v}_1, \vec{v_2}, \vec{c}_s) = F((\vec{v}_1 - \vec{v}_2)^T \vec{c}_s) = F(\vec{v}_1.\vec{c}_s) \otimes_q F(\vec{v}_2.\vec{c}_s)^{-1}, \tag{8}$$

where $\vec{v}_1, \vec{v_2}, \vec{c}_s$ represent vectors of two words and the discourse vector, respectively; $D_q$ is the set of the functions $F$ and $\otimes_q$ is a generalization of the product, which is called the q-product, (Nivanen et al., 2003). $\otimes_q$ is defined as follows:

$$x \otimes_q y = sign(x)sign(y)[x^{1-q} + y^{1-q} - 1]^{\frac{1}{1-q}} \tag{9}$$

The solution to Equation 8 is the q-exponential function, $exp_q(x) = [1 + (1 - q)x]^{1/(1-q)}$. One recovers the log-linear model in the case of extensivity, i.e. $q \to 1$.

## 3.1 THE LIKELIHOOD FUNCTION

By adopting the q-exponential function as a generative model we rewrite the likelihood of having a word $w$ emitted in a sentence $s$:

$$Pr(w|s) = \alpha P(w) + (1 - \alpha)\frac{exp_q(\vec{v}_w.\tilde{c}_s)}{Z_q}, \tag{10}$$

$Z_q$ is the non-extensive partition function defined by $Z_q = \Sigma_{v_w} exp_q(\vec{v}_w.\tilde{c}_s)$. In our work we make the assumption of homogeneity and assume that the partition function is constant, $Z_q = Z$. In this case, the discourse vector $\tilde{c}_s$ is approximated to a non-linear average using the $\oplus_q$ algebra. This means that the discourse vector will take into account the similarity between the word vectors and add more "attention" to the direction dictated by similar words. More particularly, the the q-sum between two vectors $v_1$ and $v_2$ as:

$$v_1 \oplus_q v_2 = v_1 + v_2 + (1 - q)v_1 \odot v_2, \tag{11}$$

where $\odot$ is the Hadamard product. The term proportional to the Hadamard product represents the non-linearity of the q-sum and conveys information about the similarity between the two vectors.

## 3.2 LIFTING THE CONFOUNDING EFFECT OF VECTOR LENGTH

In this section, we show how confounding effect of the word vector length is mitigated when using the non-extensive algebra.
Given the same assumptions, the previous discourse vectors in Section 2.2 are given by:

$$c_g = v_x \oplus_q v_y = v_x + v_y, c_h = v_z \oplus_q v_z = v_z + v_z + (1 - q)[k^2 e_x + k^2 e_y]. \tag{12}$$

Taking the inner product of the discourse and word vectors, we have:

$$< c_g, x > + < c_g, y > \le 2 < c_h, z > \implies p(g|c_g) \le p(h|c_h) \tag{13}$$

and thus there is no confounding effect of vector length in the non-extensive algebra framework in the case where the lengths of the vectors are of the same order.

## 3.3 DERIVING THE SENTENCE EMBEDDING

In the following we maximize the log-likelihood and derive the algorithm for the sentence embedding. The likelihood of a sentence $s$ could be written as:

$$p[s|c_s] = \Pi_{w \in s}p(w|c_s) = \Pi_{w \in s}[\alpha P(w) + (1 - \alpha)\frac{exp_q(\vec{v}_w.\tilde{c}_s)}{Z}] \tag{14}$$

Considering the q-log-likelihood:

$$f_w(\tilde{c}_s) \approx log_q[\alpha P(w) + (1 - \alpha)\frac{exp_q(\vec{v}_w.\tilde{c}_s)}{Z}], \tag{15}$$

Using the property

$$\frac{dexp_q(x)}{dx} = exp(x)^q, \tag{16}$$

we compute the derivative:

$$\Delta f_w(\tilde{c}_s) = \frac{(1 - \alpha)(exp_q(\tilde{c}_s\vec{v}_w))^q}{(\alpha p(w) + (1 - \alpha)exp_q(\tilde{c}_s.\vec{v}_w)/Z)^q Z}v_w \tag{17}$$

Then, by performing a Taylor expansion, we get:

$$f_w(\tilde{c}_s) \approx f_w(0) + \Delta f_w(0)^T c_s$$
$$= constant + \frac{(1 - \alpha)}{(\alpha p(w) + (1 - \alpha)/Z)^q Z} < v_w, c_s > \tag{18}$$

Therefore, the maximum likelihood estimator for $c_s$ on the unit sphere ignoring normalization is,

$$argmax\Sigma_{w \in s}f_w(\tilde{c}_s) \propto \Sigma_{w \in s}\frac{(1 - \alpha)}{(\alpha p(w) + (1 - \alpha)/Z)^q Z}v_w, \tag{19}$$

where $\Sigma_{w \in s}$ represent the q-sum over the word vectors in the sentence.

---

**Algorithm 1** Sentence Embedding

---

1: **Input:** Word embedding $\{v_w : w \in \mathbf{V}\}$, a set of sentences $S$, parameter $a$ and estimated probabilities of word occurrences $\{p(w) : w \in \mathbf{V}\}$ of words.
  **Output:** Sentence embeddings $\{v_s : s \in \mathbf{S}\}$
2: **for all** Sentence s in S **do**
3:   $v_s \leftarrow \frac{1}{|s|}\Sigma_{w \in s} \frac{(1-\alpha)}{Z(\alpha p(w)+(1-\alpha)/Z)^q}$
5: **end for**
   Construct a matrix M whose columns are $v_s : s \in S$, and let $u$ be its first singular vector
6:   **for all** sentence s in $S$ **do**
7:     $v_s \leftarrow v_s - uu^T v_s$
8:   **end for**

---

# 4 EXPERIMENT

## 4.1 TEXTUAL SIMILARITY TASK

We test our method on the SemEval semantic textual similarity (STS) tasks (Agirre et al., 2013; 2014; 2015). The goal of this task is to quantify the similarity between two different sentences. To compute the similarity of a sentence pair, we embed the sentences and compute cosine distance between the two corresponding vectors. The word frequency $p(w)$ is estimated on the Wikipedia corpus (**enwik8**). We chose the weighting parameter $a = 10^{-1}$ and the partition function $Z = 10^3$. The extensivity parameter is taken as $q = 0.7$. This value are empirically chosen based on the optimization of the textual similarity score (a more systematic optimization is left for future work)
We compare the results for textual similarity obtained from our model with those from several other methods, which are categorized by (Cer et al., 2017) as unsupervised, weakly supervised, or supervised.

The performance on the textual similarity task is quantified using the Pearson's correlation coefficient estimated between the reference similarity and the cosine distance obtained with our model.

## 4.2 EXPERIMENTAL SETTINGS

In our experiment, we try three word embedding methods: GloVe vectors (Pennington et al., 2014b), trained on both Twitter and Wikipedia corpora, PARAGRAM-SL999 (PSL) vectors (Wieting et al., 2015), and ParaNMT-50 vectors (Wieting & Gimpel, 2017), based on 51M EnglishEnglish sentence pairs translated from EnglishCzech sentence pairs.

## 4.3 RESULTS

We illustrate the results in Table 1. It is shown that our model gives comparative results with (Ethayarajh, 2018) and outperforms the model proposed by (Arora et al., 2016). Best performances in terms of similarity score are obtained using the Glove (Wikipedia) embedding and the ParaNMT embedding. On average best performance in similarity tasks is obtained using the ParaNMT embedding.
The performance of the model drops drastically as we converge towards the extensive case, $q \to 1$, with on average nearly $20\%$ drop in performance in terms of similarity.

# 5 CONCLUSION AND FUTURE WORK

In this short paper, we propose a new approach for sentence embedding based on non-extensive statistics. We argue that due to the complex nature of semantics, highly non-linear behaviour in the embedding space have to be taken into account. For example, the discourse vector is integrated as being proportional to the Hadamard product, accentuating the semantic similarity between the vectors integrated over in a non-linear fashion. In addition, it has been shown that by considering the q-sum, the compounding effect of the vector length is mitigated under certain conditions. However, we do not drop the dependence of the likelihood on the vector length, as we believe that doing so will lead to a loss of semantic information.

Table 1: Experimental results on textual similarities in tems of Pearson's correlation coefficient (Pearsons r  100). Our method is applied using three embedding methods: Glove, PSL, and ParaNMT. The highest score in each column is in highlighted.

| Model | STS'12 | STS'13 | STS'14 | STS'15 |
|---|---|---|---|---|
| Wieting et al. (2016b) - unsupervised | | | | |
| PP | 58.7 | 55.8 | 70.8 | 75.8 |
| tfidf-Glove | 58.7 | 52.1 | 63.8 | 60.6 |
| PP-XXL | 61.5 | 58.9 | 73.1 | 77.0 |
| Wieting et al.(2017b) - weakly supervise | | | | |
| LSTM AVG | 64.8 | 63.1 | 75.8 | 76.7 |
| AVG | 61.6 | 59.4 | 75.8 | 77.9 |
| GRAN | 62.5 | 63.4 | 75.9 | 77.7 |
| Conneau et al. (2017) - unsupervised (transfer learning) | | | | |
| InferSent (AllSNLI) | 58.6 | 51.5 | 67.8 | 68.0 |
| InferSent (SNLI) | 57.1 | 50.4 | 66.2 | |
| Arora et al. (2017) - weakly supervise | | | | |
| GloVe+WR | 56.2 | 56.6 | 68.5 | 71.7 |
| PSL+WR | 59.5 | 61.8 | 73.5 | 76.3 |
| Ethayarajh et al. (2018) | | | | |
| GloVe+UP | 64.9 | 63.6 | 74.4 | 76.1 |
| PSL+UP | 65.8 | 65.2 | 75.9 | 77.6 |
| ParaNMT+UP | 68.3 | **66.1** | 78.4 | **79.0** |
| Our approach | | | | |
| GloVe (Wikipedia) | 66.8 | 60.2 | **78.1** | 78.5 |
| GloVe (Twitter) | 66.3 | 61.2 | 77.5 | 77.2 |
| PSL | 67.1 | 65.3 | 75.5 | 73.1 |
| ParaNMT | **71.4** | 65.2 | 77.7 | 76.2 |

The proposed method is weakly supervised since one needs to determine the parameter $\alpha$ and the extensivity index $q$. In future work, we will aim to impose constraints on the system in order to systematically determine the degree of extensivity and hence the value of the parameter q. In addition, the assumption of the homogeinity of the word vectors in the embedding space, and hence the assumption of an invariant partition function needs to be investigated further.

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
