# OpenReview forum: "A NON-LINEAR  THEORY FOR SENTENCE EMBEDDING"
_ICLR.cc/2019/Conference_

### Official Review · AnonReviewer3 · 2018-11-01
**This paper claims to introduce non linearity in the discourse vector framework defined by Arora et.al. While the motivations for the non extensive statistics seem interesting and warrant a thought, the experiments are severely lacking in providing any insight into the method.**

**Rating:** 3
**Confidence:** 4

**Review:**

This paper while presenting interesting ideas, is very poorly written. It seems as though the authors were in a rush to submit a manuscript and did not even bother with basic typesetting.
Firstly, the paper spends too much time motivating and re-introducing the model of Arora et.al. Note to the authors here, they cite the same paper from Arora et.al for 2017 twice. The first time the model they refer to was introduced by the paper "RAND-WALK: A latent variable model approach to word embeddings", this is probably what the authors mean by the 2016 reference?

Now coming to the experiments, the results are presented in a table that is poorly formatted. The section partitions are not clearly delimited, making for a hard read. Even if we overcome that and look at the results, the presented numbers are incredibly confusing. On the STS 13 and 15 data sets, Ethayarajh 2018's numbers are much better at 66.1 and 79.0. Coming to STS14 Ethayarajh attain 78.4 while the proposed method achieves 78.1. If we discount this for the moment, and look at the results on STS12 where the proposed method achieves 71.4, this is the only data set where the proposed method does better than the other baselines.

So almost on 3 of the 4 datasets Ethayarajh 2018 does better. This makes me question what exactly is the proposed model improving?

Coupled with the fact that there is no motivation to explain results or future work, this makes for a very poorly written paper that is very challenging to read.

It is very likely that there is some merit to the proposed methods that introduce non linearity, but these points simply get lost in the mediocre presentation.

---

### Official Review · AnonReviewer1 · 2018-11-01
**This paper is not positioned well with respect to the literature. I am not sure what are its key contributions and how significant they are.  The technical exposition also appears somewhat incoherent and not well-justified (see my specific comments below).**

**Rating:** 3
**Confidence:** 3

**Review:**

PAPER SUMMARY:

This paper introduces a non-extensive statistic random walk model to generate sentence embedding while accounting for
high non-linearity in the semantic space.

NOVELTY & SIGNIFICANCE:

I am not sure what the main focus of this paper is. It seems accounting for non-linearity in the semantic space while generating sentence embedding has already been achieved by existing LSTM models -- the goal seems to be more about interpretability and computational efficiency but the paper did not really discuss these in detail (more on this later).

In terms of the proposed solution, I am also not sure what is the significance of using non-extensive statistic in this context. In fact, the background section gave the impression that the non-linear form of q-exponential is the main reason to advocate this approach. But, if it is only about handling non-linearity, there are plenty of alternatives and it is important to point out exactly what advantages non-extensive statistic has over the existing literature (e.g., why is it more interpretable than LSTM). Please expand the respective background section to clarify this.

TECHNICAL SOUNDNESS:

There are parts of the technical exposition that appear confusing and somewhat incoherent. For instance, what exactly is this confounding effect of vector length & why do we need to address this issue if according to the Section 2.2, it has already been addressed in the same context?

Section 2.2 seems to discuss this effect but the exposition is unclear to me. The authors start with an example and a bunch of assumptions that lead to a contradiction.

It is then concluded that the cause of this is due to the linearity assumption (what about the other assumptions?) in estimating the discourse vector.

I do not really follow this reasoning and it would be good if the authors can elaborate more on this.

CLARITY:

The paper seems to focus too much on technical details and does not give enough discussion on its positioning. The significance of the proposed solution with respect to the literature remains unclear.

EMPIRICAL RESULTS:

I am not an expert in this field and cannot really judge the significance of the reported results. I do, however, have a few questions: in all benchmarks, are the algorithms tested on a different domain than the domain it was trained on?

Have the authors compared the proposed sentence embedding framework with the LSTM literature mentioned in the introduction? I noticed there was a LSTM AVG in the comparison table.

Is that the simple averaging scheme mentioned in the introduction when the authors discussed transferrable sentence embedding?

Is there any reason for not comparing with RNN (Cho et al., 2014)?

In terms of the computation processing cost, how efficient is the proposed method (as compared to existing literature)?

---

### Official Review · AnonReviewer2 · 2018-11-06
**Insufficient contents and experiments. Not ready to be published.**

**Rating:** 3
**Confidence:** 3

**Review:**

Summary: this paper discussed an incremental improvement over the Random Walk based model for sentence embedding.

Conclusion: this paper is not ready for publication, very poor written and well below the bar of ICLR-caliber papers.

More:
This paper spent the majority of its content explaining background (those paragraphs were very poor written and difficult to read), and very briefly introduced their methodology with some mathematical derivations and equations, most of which can be put in the supplement instead of main context. The author didn't quite explain how the proposed method, such as why using non-extensive statistic in this context,

The experiment results aren't convincing and lack sufficient information for reproducibility.

---

### Meta-Review · Area_Chair1 · 2018-12-08

**Confidence:** 4
**Recommendation:** Reject

**Metareview:**

The paper is poorly written and below the bar of ICLR. The paper could be improved with better exposition and stronger experiment results (or clearer exposition of the experimental results.)

---

> ### Author Response · Authors · 2019-01-22
> **Thank you for your helpful feedback**
>
> We thank the reviewers for providing productive comments and critiques. We believe this input is very useful. We are working on improving the presentation of the paper as well as extending the non-extensive theory to several other applications. We will present our enhanced model in future opportunities.